# FailureAtlas: Mapping the Failure Landscape of T2I Models via Active Exploration

## Abstract

Static benchmarks have provided a valuable foundation for comparing Text-to-Image (T2I) models. However, their passive design offers limited diagnostic power, struggling to uncover the full landscape of systematic failures or isolate their root causes. We argue for a complementary paradigm: **active exploration**, and introduce **FailureAtlas**, the first framework designed to autonomously explore and map the vast failure landscape of T2I models at scale. FailureAtlas frames error discovery as a structured search for minimal, failure-inducing concepts. While it is a computationally explosive problem, we make it tractable with novel acceleration techniques. When applied to Stable Diffusion models, our method uncovers hundreds of thousands of previously unknown error slices (over 247,000 in SD1.5 alone) and provides the first large-scale evidence linking these failures to data scarcity in the training set. By providing a principled and scalable engine for deep model auditing, FailureAtlas establishes a new, diagnostic-first methodology to guide the development of more robust generative AI.

## 1 Introduction

Static benchmarks have been invaluable for tracking the progress of Text-to-Image (T2I) models, providing a common ground for comparison (Rombach et al., 2022; Saharia et al., 2022). However, this evaluation paradigm, inherited from discriminative AI, is fundamentally limited. Relying on fixed, often imbalanced prompt sets, these *passive* methods offer shallow diagnostic power; they reveal that a model fails, but struggle to isolate *why*, often conflating multiple error-causing factors (Hu et al., 2023; Huang et al., 2023). As a result, our understanding of T2I model weaknesses remains superficial, hindering targeted improvements.

To overcome these limitations, we argue for a shift towards a complementary paradigm: **active exploration**. Rather than passively grading models on a fixed test set, we propose to actively and systematically probe their capabilities to create a comprehensive map of their failure landscape. This approach enables the precise localization of minimal, fundamental attributes that trigger generation failures, offering a deeper, more actionable form of model analysis. However, making this paradigm a reality presents two formidable challenges: the vast, combinatorial nature of the T2I input space and the prohibitive computational cost of image generation.

In this paper, we introduce FailureAtlas, the first framework to make large-scale active exploration practical. We address the challenge of the infinite search space by first constructing a massive, high-coverage entity-attribute corpus, which structures the problem. We then navigate this space with a systematic tree search. To tackle the prohibitive cost, we introduce two powerful acceleration strategies: a monotonic rule-based pruning to eliminate redundant exploration, and a novel prediction-based prioritizer that learns the model's error distribution to focus the search, significantly improving the efficiency of failure discovery. An overview is shown in Figure 1.

By adopting the active paradigm, our approach yields diagnostic insights unattainable by traditional methods. FailureAtlas automatically uncovers over 247,000 fine-grained error slices in SD1.5 (Rombach et al., 2022) and 439,000 in SDXL Turbo (Podell et al., 2023), revealing detailed capability profiles of each model. Furthermore, by aligning error slices with the training data distribution through our corpus, we provide the first large-scale evidence linking these discovered failures back to their potential origins, such as data scarcity in the LAION-2B dataset (Schuhmann et al., 2022). Our contributions are threefold:

Figure 1: FAILUREATLAS actively explores and maps the failure landscape of T2I models. It frames error discovery as structured tree search accelerated by pruning and prioritization, and finds minimal, failure-inducing concepts.

1. We introduce **the first active exploration framework for T2I models**, shifting from static scoring to systematic, large-scale diagnostic mapping.

2. We operationalize this paradigm with a **high-coverage corpus and structured tree search**, and ensure scalability with **rule-based and prediction-based acceleration strategies**, enabling the efficient discovery of minimal, fundamental error slices at scale.

3. We provide a **scalable analysis method** that leverages our corpus as a bridge to link T2I failures to their potential origins.

## 2 RELATED WORKS

### 2.1 LIMITATION OF BENCHMARK-DRIVEN EVALUATION

The evaluation of text-to-image (T2I) generative models are predominantly benchmark-driven (Lin et al., 2014). A range of benchmarks has been introduced to probe different dimensions of T2I performance. For instance, GenEval (Ghosh et al., 2023) emphasizes object- and attribute-level alignment (e.g., color, count, position), while T2I-CompBench (Huang et al., 2023) extends the scope to compositional relations and texture. More recent benchmarks, such as HRS-Bench (Bakr et al., 2023) and WISE (Niu et al., 2025), further assess high-level reasoning, commonsense, and world knowledge. In parallel, automatic evaluation metrics have also been developed. Early metrics such as CLIPScore (Hessel et al., 2021) and PickScore (Kirstain et al., 2023) capture global text-image consistency, whereas more recent ones (e.g., TIFA (Hu et al., 2023), VQAScore (Lin et al., 2024)) enable finer-grained assessments of attribute alignment. Together, these benchmarks and metrics have substantially advanced the evaluation of T2I systems.

However, existing efforts largely focus on cross-model comparison and aggregate performance assessment, offering limited insight into systematic error patterns. Their prompt distributions are often imbalanced, resulting in uneven and incomplete coverage across concepts and attributes. Moreover, failures identified through benchmarks frequently conflate multiple attributes, making it difficult to isolate the specific factors responsible. Consequently, the challenge of systematically discovering, characterizing, and analyzing model errors remains insufficiently addressed in current T2I evaluation practices.

### 2.2 ERROR SLICE DISCOVERY

Error slice discovery, which aims to uncover systematic model errors, has been extensively explored in discriminative models. Early approaches cluster failure cases in embedding spaces (e.g.,

CLIP (Radford et al., 2021)) and annotate clusters with natural language descriptionsEyuboglu et al. (2022); Yenamandra et al. (2023), though the resulting slices often suffer from poor coherence and interpretability due to entangled representations (Chen et al., 2024). More recent work adopts a *tag-then-slice* paradigm (Gao et al., 2023; Chen et al., 2024; 2025), which first generates task-related attributes or tags—via human inspection or multi-modal LLMs—and then groups data into slices accordingly. These techniques enable failure analysis but remain bounded by passive evaluation datasets, and thus suffer from imbalanced test coverage and difficulty in isolating the precise factors driving failures.

Extending error slice discovery to generative models introduces both opportunities and challenges. For T2I models, inputs can be flexibly constructed with fine-grained control, enabling active probing through tailored prompts rather than relying on fixed datasets. In this active setting, one can exercise finer control over the granularity, order, and scope of exploration, which supports broad and balanced coverage across concepts and the isolation of minimal attributes that drive failures. At the same time, generative models pose new difficulties: their input prompt space is inherently vast, and each evaluation is computationally expensive due to slow image generation. Taken together, these challenges underscore the need for dedicated error slice discovery methods for T2I generation. In this paper, we propose FAILUREATLAS, which is the first active exploration framework for error slice discovery in T2I models.

## 3 METHODS

FAILUREATLAS discovers error slices by leveraging an entity–attribute corpus to define a broad search space (Sec 3.1) and applying a structured tree search (Sec 3.2) with rule-based and prediction-based accelerations (Sec 3.3) to discover errors. With the model's training data, we can additionally analyze slice-level data distributions to provide further context for interpreting errors (Sec 3.4).

### 3.1 STRUCTURING THE SEARCH SPACE: THE ENTITY–ATTRIBUTE CORPUS

The corpus defines the structured search space for error slice discovery and consists of two complementary components: an entity corpus and an attribute corpus. By composing entities with attributes, we can generate a large and diverse set of model input prompts. The construction of the corpus involves three stages.

First, we initialize a base vocabulary using a large language model (LLM), which generates entities and attributes from general world knowledge to ensure broad coverage across diverse object categories, scenes, and visual properties. Second, we refine and expand this vocabulary through large-scale dataset mining. Entities and attributes are extracted from two representative datasets—COCO Captions (Lin et al., 2014) and T2I-CompBench (Huang et al., 2023). Extracted terms are aligned with the base vocabulary, while frequently occurring but unmatched terms are added as new entries. The vocabulary is then iteratively curated to remove redundancy, and establish hierarchical categories (e.g., "biology" → "animal" → "dog") for both entities and attributes. To balance corpus size with semantic coverage, the entity corpus emphasizes general terms while deliberately avoiding proper nouns or fine-grained concepts (e.g., general term tower vs. fine-grained concepts such as Eiffel Tower or Tokyo Tower).

Finally, we use an LLM to annotate the semantic validity of each entity–attribute pair, ensuring that the search process avoids generating implausible prompts. The resulting corpus contains 758 entities and 437 attributes, providing a structured yet flexible foundation for systematic error exploration. An overview is presented in Figure 2.

### 3.2 ERROR SLICE DISCOVERY

Using this corpus as foundation, as shown in Figure 3, we formalize error slice discovery as a layered combinatorial search over the entity–attribute space, represented as a tree structure. Each node in the tree corresponds to a specific entity–attribute combination: the root layer contains entities only; the second layer adds a single attribute, and deeper layers progressively introduce additional attributes. The search proceeds in a breadth-first manner across multiple entity trees, exploring all nodes at a given depth before moving to the next layer. To constrain the search space, we restrict each search

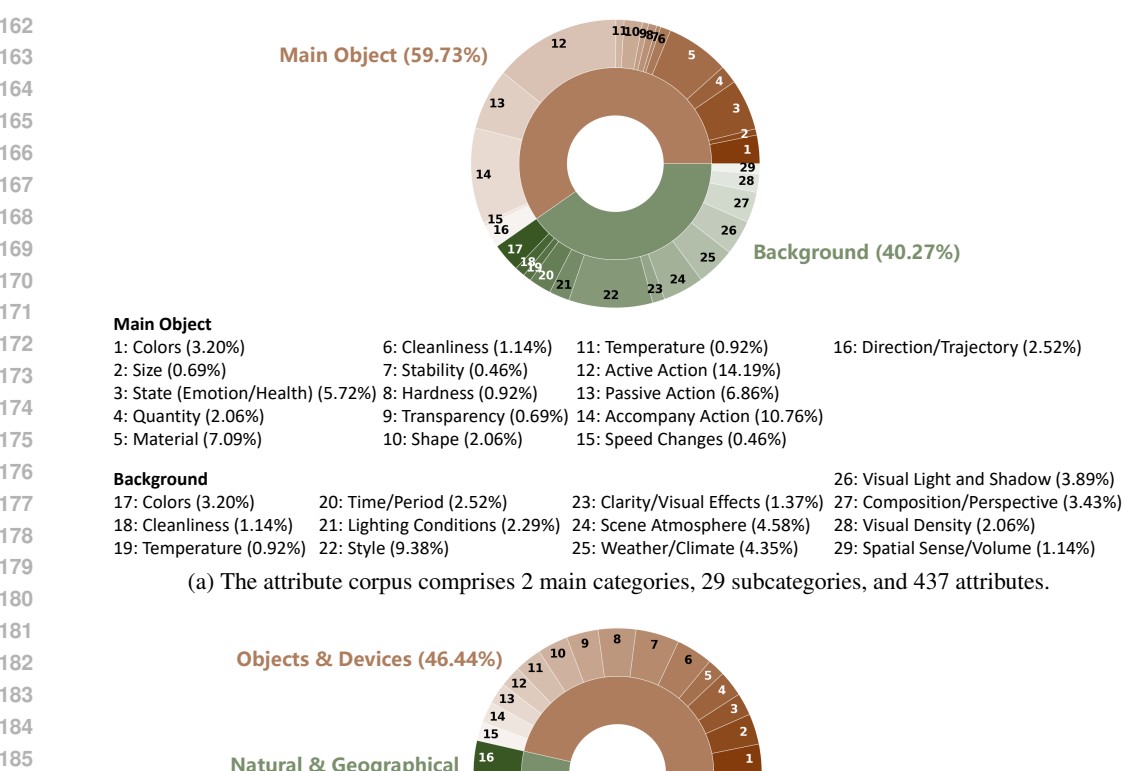

**Main Object**

| | | | |
|---|---|---|---|
| 1: Colors (3.20%) | 6: Cleanliness (1.14%) | 11: Temperature (0.92%) | 16: Direction/Trajectory (2.52%) |
| 2: Size (0.69%) | 7: Stability (0.46%) | 12: Active Action (14.19%) | |
| 3: State (Emotion/Health) (5.72%) | 8: Hardness (0.92%) | 13: Passive Action (6.86%) | |
| 4: Quantity (2.06%) | 9: Transparency (0.69%) | 14: Accompany Action (10.76%) | |
| 5: Material (7.09%) | 10: Shape (2.06%) | 15: Speed Changes (0.46%) | |

**Background**

| | | | |
|---|---|---|---|
| | | | 26: Visual Light and Shadow (3.89%) |
| 17: Colors (3.20%) | 20: Time/Period (2.52%) | 23: Clarity/Visual Effects (1.37%) | 27: Composition/Perspective (3.43%) |
| 18: Cleanliness (1.14%) | 21: Lighting Conditions (2.29%) | 24: Scene Atmosphere (4.58%) | 28: Visual Density (2.06%) |
| 19: Temperature (0.92%) | 22: Style (9.38%) | 25: Weather/Climate (4.35%) | 29: Spatial Sense/Volume (1.14%) |

(a) The attribute corpus comprises 2 main categories, 29 subcategories, and 437 attributes.

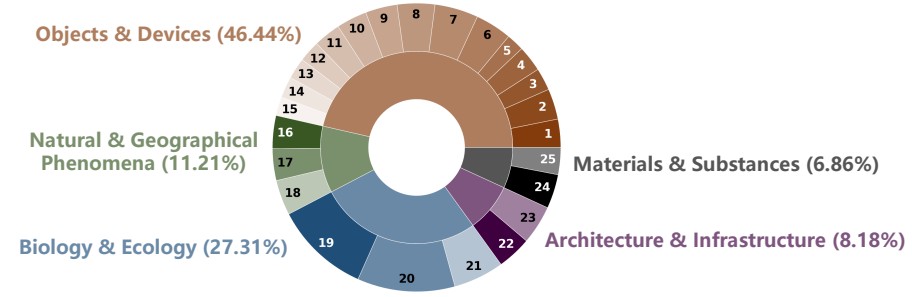

**Objects & Devices**

| | | |
|---|---|---|
| 1: Tools & Accessories (3.17%) | 6: Goods & Consumer Products (4.09%) | 11: Outdoor Facilities (3.03%) |
| 2: Home Appliances & Electronics (3.43%) | 7: Food & Beverages (4.88%) | 12: Weapons & Protective Gear (2.37%) |
| 3: Transportation (2.51%) | 8: Daily Household Items (4.22%) | 13: Religious & Ritual Items (2.37%) |
| 4: Furniture & Decorations (2.90%) | 9: Sports & Sporting Equipment (3.56%) | 14: Culture & Art (2.37%) |
| 5: Machinery & Equipment (1.98%) | 10: Medical & Health Equipment (3.43%) | 15: Antiquities & Relics (2.11%) |

| **Natural & Geographical Phenomena** | **Biology & Ecology** | **Architecture & Infrastructure** | **Materials & Substances** |
|---|---|---|---|
| 16: Geographical Features (3.69%) | 19: Animals (10.69%) | 22: Buildings & Structures (3.83%) | 24: Natural Materials (3.83%) |
| 17: Natural Phenomena (3.56%) | 20: Plants (10.95%) | 23: Infrastructure (4.35%) | 25: Synthetic Materials (3.03%) |
| 18: Celestial Bodies & Astronomy (3.96%) | 21: Others (5.67%) | | |

(b) The entity corpus comprises 5 main categories, 25 subcategories, and 758 entities.

Figure 2: Statistics of the entity-attribute corpus.

path to at most one attribute from a category (e.g., if "big" is chosen from "size", "small" cannot be added along the same path).

For each node, we construct a prompt by inserting the entity and its attributes into a predefined template (details in Appendix A.4.1). Then, we use an LLM to make only minor grammatical corrections. This two-step construction minimizes complexity and avoids uncontrolled expansion. Each prompt is then used to generate multiple images with the target T2I model (e.g., 25 images per node).

The generated images are evaluated with a multi-choice consistency check. For each attribute or entity in the corpus, we construct a set of semantically related alternatives by words in the same category and present them as candidate answers (details in Appendix A.4.2) to a multimodal large language model (MLLM). The MLLM selects the closest description based on the image. Given multiple generated images for a node, each evaluated on attributes and entity, we define the generation success rate as the percentage of MLLM predictions that align with the intended attributes or entity. A node is classified as an error slice if its success rate falls below a threshold $\tau$.

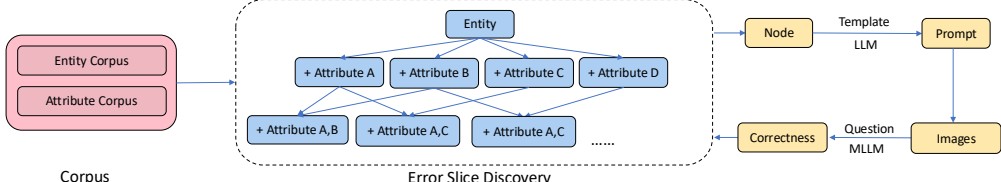

Figure 3: Error slices are discovered via structured tree search over combinations of entities and attributes. Each node of the tree is converted into a prompt by an LLM, used to generate images with the T2I model, and evaluated for correctness by a MLLM. This process enables automatic identification of fine-grained failure cases.

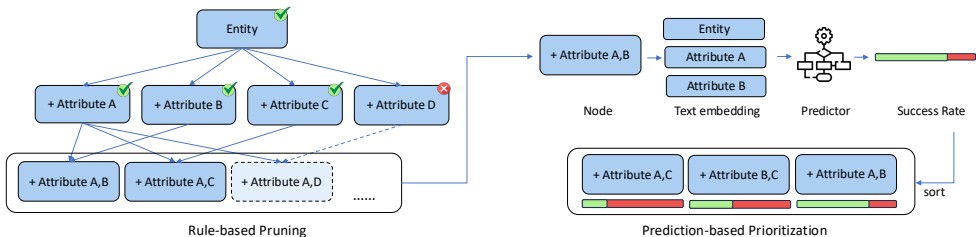

Figure 4: Acceleration strategies for error slice discovery. Left: Rule-based pruning skips deeper nodes once a parent fails. Right: Prediction-based prioritization uses a learned predictor to prioritize nodes with higher error likelihood.

### 3.3 MAKING EXPLORATION TRACTABLE: ACCELERATION STRATEGIES

**Rule-based pruning.** Due to the combinatorial explosion of the search space, a brute-force tree search is computationally infeasible. To address this, as shown in Figure 4, we apply two acceleration strategies to efficiently discover errors. Since our search procedure is tree-structured, we apply pruning to reduce the search space. In text to image generation, generating images binding with multiple attributes is generally more challenging than with its attributes subset. We assume that difficulty increases along each search path. Consequently, if a node already fails (e.g., "dog" fails), all of its deeper extensions that add more attributes (e.g., "jumping dog") are skipped. This not only improves efficiency but also ensures that the discovered errors correspond to fundamental failures, i.e., the minimum concepts driving the error.

While this assumption generally holds, there are rare cases where adding attributes reduces ambiguity and thus makes generation easier (e.g., "a strawberry" vs. "a red strawberry"). Such exceptions, however, are uncommon and have negligible impact on our analysis. More importantly, since our objective is to identify the fundamental failure concepts, a failure at a parent node remains informative: even if one of its deeper extensions were to succeed, the parent's failure still exposes the minimal concept the model cannot properly handle.

**Prediction-based prioritization.** The tree search procedure is conducted layer-by-layer. While pruning introduces dependencies between nodes across layers, nodes within the same layer remain independent, allowing their exploration order to be flexibly adjusted. Empirically, we observe strong intra-entity attribute correlations: for instance, if a T2I model fails on one large-motion + entity combination, it often fails on other large-motion combinations involving the same entity. This suggests that a node's success rate can be inferred from the outcomes of related nodes.

To leverage this property, we train a lightweight predictor that takes as input the text embeddings of entities and their attributes from a pretrained encoder, and outputs the estimated success rate of each node. The predictor is implemented as a transformer decoder, where depth-varying embeddings are padded with zeros at lower layers, and multiple cross-attention blocks are applied between entity and attribute embeddings. During error slice discovery, the model is updated online with newly obtained evaluation results and guides the prioritization of nodes with a high predicted likelihood of failure (details in Appendix A.4.3). This prioritization strategy allows the framework to uncover a large number of errors within limited computational budgets.

### 3.4 FROM DISCOVERY TO DIAGNOSIS: ERROR ATTRIBUTION

We focus on data scarcity as a potential factor contributing to generation errors. To analyze this factor, we first extract entities and attributes from the model's training set and align them with our corpus vocabulary. Based on the layer structure defined in error slice discovery, we compute the average data distribution of slices at each layer of the search tree, where distribution is measured as the proportion of training samples containing the slice. For each discovered error slice, if the occurrence frequency falls below a threshold given by $\alpha$-scaled layer-wise average, we attribute the error to *data scarcity*. Such attribution does not imply that simply adding more data will fully resolve the failure, but rather provides indicative signals for targeted dataset curation and model improvement.

## 4 EXPERIMENTS

### 4.1 EXPERIMENTAL SETUP

To assess the effectiveness and efficiency of FAILUREATLAS, we design experiments along three key dimensions: (i) semantic coverage of the entity–attribute corpus, (ii) performance of acceleration strategies, and (iii) error slice discovery and attribution capability.

We evaluate FAILUREATLAS on two representative T2I models: Stable Diffusion 1.5 (SD1.5) (Rombach et al., 2022) and Stable Diffusion XL Turbo (SDXL Turbo) (Podell et al., 2023). To balance computational feasibility and semantic coverage, we restrict the search depth to three layers, meaning each entity can be combined with at most two attributes. This design captures a broad range of entity–attribute bindings while keeping the search space tractable. The resulting potential search space contains approximately 7.36M nodes. For each node, we generate 25 images using the target T2I model. The MLLM for automatic evaluation is Qwen2-VL-72B (Wang et al., 2024). In error slice discovery, prediction-based prioritization leverages T5 (Raffel et al., 2020) to generate text embeddings. The threshold $\tau$ of generation success rate is set 0.8. This setting is consistently used across all experiments. We present more implementation details in Appendix A.5.1.

### 4.2 SEMANTIC COVERAGE OF THE ENTITY–ATTRIBUTE CORPUS

We begin by assessing the semantic coverage of our entity and attribute corpus to common scenarios in T2I generation. To this end, we extract entities and attributes from two widely used T2I benchmarks—HRS-Bench (Bakr et al., 2023) and TIFA (Hu et al., 2023)—and compute the proportion that can be mapped to our vocabulary. For reference, we also report results on COCO Captions (Lin et al., 2014) and T2I-CompBench (Huang et al., 2023), which served as sources during corpus construction. As shown in Table 1, our corpus covers around 90% of both entities and attributes across all four benchmarks, highlighting its semantic representativeness and its suitability for systematic error slice discovery.

Table 1: Semantic coverage of entities and attributes across four benchmarks.

| Benchmark | Entity Coverage | Attribute Coverage |
|---|---|---|
| COCO Captions | 88.2% | 93.7% |
| T2I-CompBench | 88.5% | 96.0% |
| HRS-Bench | 89.6% | 90.6% |
| TIFA | 92.0% | 90.8% |

### 4.3 QUANTITATIVE IMPACT OF ACCELERATION STRATEGIES

We next evaluate the quantitative impact of the proposed acceleration strategies.

**Rule-based pruning.** This strategy avoids exploring the successors of nodes identified as failures, and its effectiveness naturally depends on the capability of the underlying models. We report the proportion of nodes pruned at each search layer across the two evaluated models. As shown in the left of Figure 5, rule-based pruning substantially reduces the search space. The reduction is more

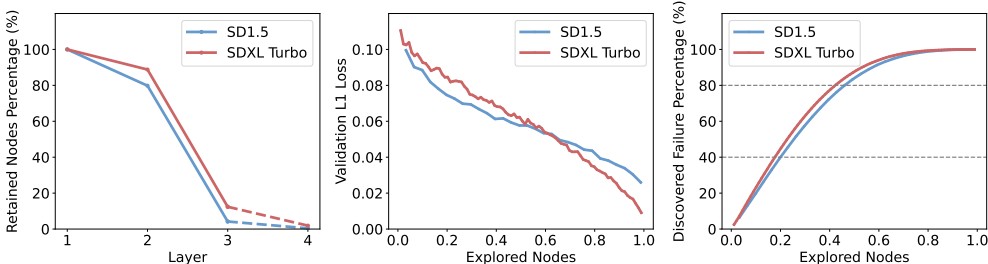

Figure 5: Left: Number of evaluated nodes compared with the full search tree. Rule-based pruning substantially reduces the search space down to 0.4% in the third layer. Middle: Validation loss (L1 score) of the online-trained predictor during search. Right: Proportion of failures discovered during search. The predictor enables roughly a 2× speed-up in failure discovery.

pronounced for weaker models and becomes increasingly significant as the number of attributes grows. For SD1.5, the search space at the third layer (entity combined with two attributes) is reduced to 4.2%, and will further shrinks to 0.4% at the extrapolated fourth layer.

**Prediction-based prioritization.** We further evaluate the prioritization model, which takes as input the text embeddings of entities and attributes of a node, and predicts the generation success rates. Based on these predictions, nodes with higher estimated error likelihood are prioritized. In our experiments, we retrain the predictor every 10,000 explored nodes and report its $L_1$ score on the subsequent 10,000 nodes (middle of Figure 5). The low $L_1$ loss on unseen nodes indicates that the model's error distribution is learnable to some extent, demonstrating the effectiveness of the predictor. We also measure the proportion of failures discovered under prioritized search in the third layer of search tree (right of Figure 5). The predictor achieves roughly a 2× speed-up in error discovery, enabling the identification of a large number of failures within limited search budgets.

Table 2: Number of error slices discovered by FAILUREATLAS. Total counts are reported for completeness but should not be directly compared across models. Lower-layer error slices correspond to more fundamental errors.

| Layer | SD1.5 | | | SDXL Turbo | | |
|---|---|---|---|---|---|---|
| | Error slices | Explored Nodes | Error Density | Error slices | Explored Nodes | Error Density |
| Layer 1 | 162 | 758 | 21.3% | 91 | 758 | 12.0% |
| Layer 2 | 113,418 | 134,816 | 84.1% | 108,111 | 150,170 | 72.0% |
| Layer 3 | 133,850 | 303,893 | 44.0% | 331,740 | 889,487 | 37.3% |
| Summary | 247,430 | 439,467 | 56.3% | 439,942 | 1,040,415 | 42.3% |

## 4.4 ERROR SLICE DISCOVERY

FAILUREATLAS discovers error slices with the minimal concepts that drive model failures. Since the search is hierarchical with pruning, weaker models fail earlier in the tree, leading to aggressive pruning and reduced ranges of explored nodes. As shown in Table 2, we explore 439,467 nodes for SD1.5 and 1,040,415 nodes for SDXL Turbo. The lower number of explored nodes shows the weaker robustness of SD1.5 relative to SDXL Turbo. In total, we discover 247,430 error slices for SD1.5 (56.3% density) and 439,942 for SDXL Turbo (42.3% density). Notably, lower-layer slices correspond to more fundamental types of errors.

Representative error slices are illustrated in Figure 6. For instance, SD1.5 fails to correctly assign colors to the background and the clock, achieving a success rate of only 57%. It also struggles to depict the action of a person bending down, with half of the generations failing. Similarly, SDXL Turbo misrepresents prompt "wooden table", where the material is rendered but the entity is missing. Even for simpler cases without attribute binding, such as "screwdriver", the model frequently fails to produce a recognizable object. To assess broader applicability, we further conduct case studies on newer models, including Flux.1 (Labs, 2025) and SD3.5 (Stability AI, 2024). We observe some interesting failure modes, such as inability to render the front view of goldfish, incorrect genera-

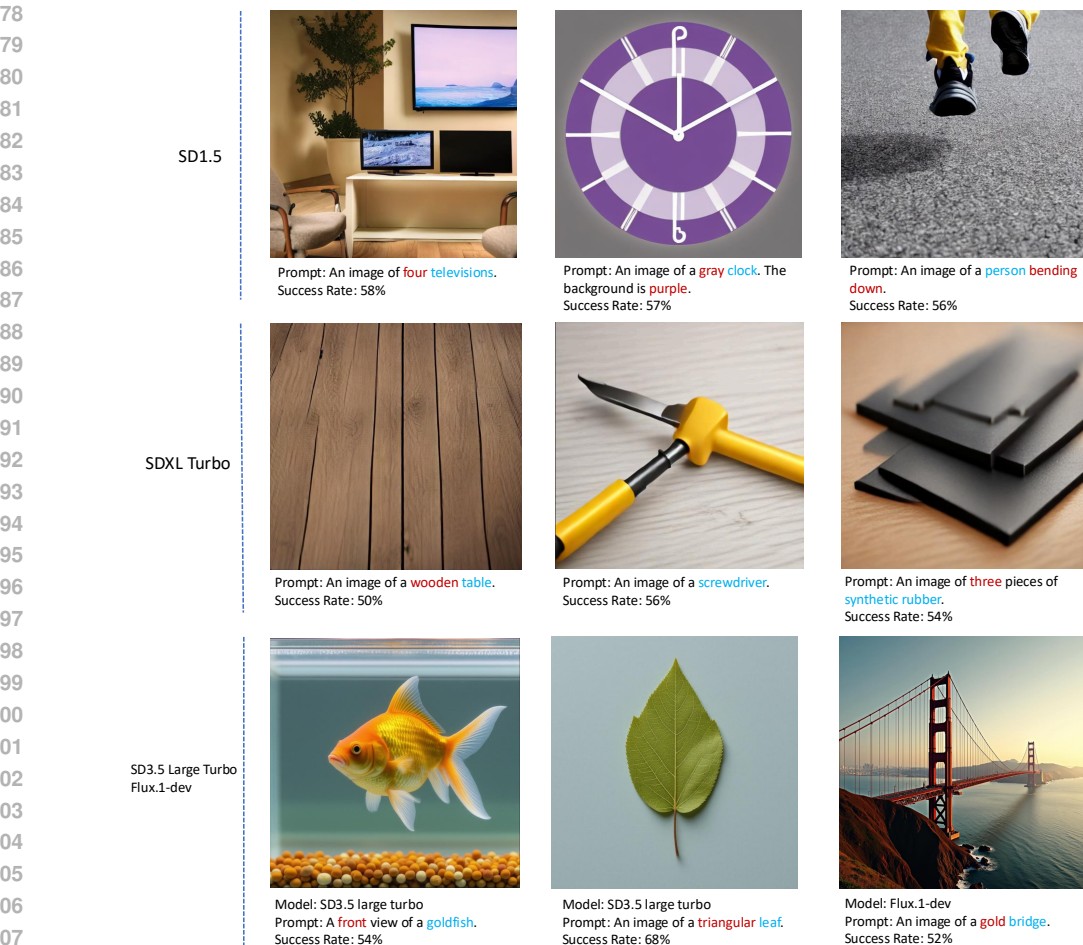

Figure 6: Example error slices of SD1.5, SDXL Turbo, SD3.5 Large Turbo and Flux.1-dev. The prompt specifies entities in blue and attributes in red.

tion of a triangular-shaped leaf, and confusion between "a gold bridge" and "the golden bridge". Visualizations and analysis of more error cases are provided in Appendix A.5.2.

## 4.5 ERROR ATTRIBUTION THROUGH DATA ANALYSIS

To explore potential causes of model errors, we conduct an auxiliary error attribution analysis for SD1.5 by examining its training data LAION-2B-en. We randomly sample 1M instances to estimate the overall distribution for computational efficiency. We summarize model performance on each entity and attribute alongside their training-set frequency in the left of Figure 7. In average, poorly performing entities or attributes coincide with insufficient training data.

We attribute error slices with occurrence frequencies lower than an $\alpha$-scaled layer-wise average to data scarcity (Section 3.4). In the right of Figure 7, we present the proportion of such slices in the first layer of tree search under varying $\alpha$ values. A substantial fraction of error slices indeed correspond to low-frequency cases, suggesting insufficient training coverage. Nevertheless, some errors occur even for relatively frequent entities ($\alpha > 0.8$), such as "badminton" and "sensor". These failures may instead arise from data quality issues, inherent generation challenges, limitations in training procedures, or architectural constraints. We leave deeper investigations of these factors to future work. While error attribution is not the main focus of this paper, these analyses provide indicative signals that can inform targeted dataset curation and model improvement.

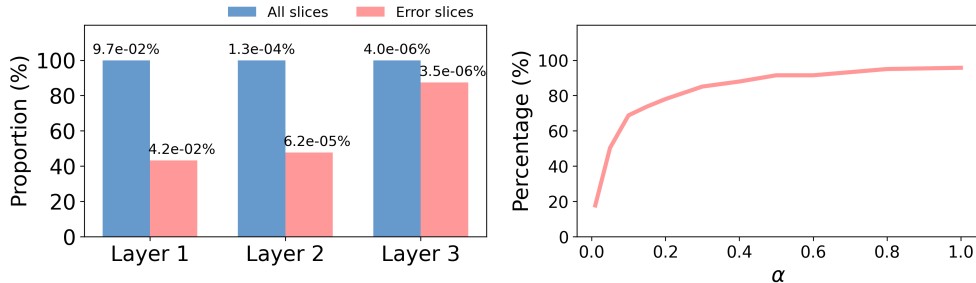

Figure 7: Left: Comparison of the average data distribution between error and total slices (normalized to 100%). Right: Proportion of error slices in the first layer under varying $\alpha$-scaled layer-wise averages. Results are reported for SD1.5 on the LAION-2B-en dataset.

## 5 DISCUSSION

**Sources of Bias.** The evaluation of a single node mainly subject to two sources of bias. The first concerns whether repeated generations of a node yield a sufficiently accurate estimate of the model's performance on a given entity–attribute pair. To examine this, we analyze how the average success rate changes with the number of repeated generations. We find that the success rate stabilizes with 25 generations, with additional samples leading to an average change of less than 0.6%. This validates the representativeness of our generation metrics. The second source of bias arises from the evaluation itself. We conduct human alignment evaluation, where the average alignment rate is 96% for entities and 84% for attributes. While text-image alignment evaluation is not the focus of this work, we expect that advances in evaluation methodology will enable increasingly precise assessments. In Appendix A.5.3, we further discuss this topic and detail the experimental settings for the above analysis.

**Scope of Error Exploration.** Our current exploration centers on the composition of a single entity with multiple attributes, without yet extending to multi-entity compositions or more complex scenarios. Despite this relatively simple design, the approach already reveals a substantial number of errors in widely used generative models. Future extensions could further deepen the exploration, particularly for key entities such as humans, which occur frequently in real-world usage.

**Method Application.** Our method is positioned as a tool for uncovering errors in the *fundamental capabilities* of T2I models. On one hand, it enables a more fine-grained analysis of model behavior and helps identify systematic capability gaps. On the other hand, by combining with data analysis, it not only guides the optimization of training data distribution and quality, but also provides a structured foundation for studying the interplay between model behavior and data characteristics.

**Time Cost.** Our method involves three computational components: generation, evaluation, and prioritization. The dominant cost comes from image generation and evaluation, which, within the same search layer, can be executed in parallel; thus, the overall runtime is determined by the slower of the two. We further discuss the time cost in Appendix A.5.4.

## 6 CONCLUSION

We introduce FAILUREATLAS, the first framework to autonomously explore and map the large-scale failure landscape of T2I models. It unifies four key components: (i) a high-coverage entity–attribute corpus that structures the search space, (ii) a layered combinatorial search procedure for error slice discovery, (iii) rule-based and prediction-based strategies that jointly accelerate exploration, and (iv) an attribution module that provides indicative data-driven insights. Together, these components enable systematic, scalable, and interpretable discovery of failure cases in T2I models. Our framework reveals fine-grained weaknesses in common T2I models and provides preliminary evidence linking these failures to data scarcity. Overall, FAILUREATLAS establishes a diagnostic-first paradigm to guide the development of more robust generative AI.

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

# A  Appendix

This appendix provides additional details to complement the main text. We elaborate on the method, including the prompt template (A.4.1), the automatic evaluation protocol (A.4.2), and the design of the prioritization model (A.4.3). We also present details of experiments introduced in the main paper, covering implementation details (A.5.1), additional visualizations and case studies (A.5.2), bias analysis (A.5.3), and a discussion of computational cost (A.5.4).

## A.1  Use of LLM

We used large language models (LLMs) to assist in writing and polishing the manuscript. LLMs were also used to suggest relevant related work during the literature review process. No part of the core methodology, data generation, analysis, or experimental design was generated or influenced by LLMs.

## A.2  Ethics statement

Our study focuses on systematic error discovery in text-to-image models through active exploration. It does not involve human subjects or private data. All models evaluated are publicly released systems, and all data used—either for training analysis or evaluation—are derived from publicly available datasets. While our work highlights potential weaknesses in generative models, it is intended solely for diagnostic and research purposes, with no intent to exploit or amplify model failures. We see this direction as a step toward more robust and interpretable generative AI systems.

## A.3  Reproducibility statement

We provide code and data to reproduce major results. Our supplementary materials include (1) code for reproduce error slices presented in the paper, and (2) the full entity–attribute corpus used in exploration. Full implementation and error slice discovery results will be released upon acceptance.

## A.4  Method Details

### A.4.1  Prompt Template

Each node is instantiated with a natural language prompt using predefined templates with three modular components:

- **Base description**, which combines entity name with quantity and descriptive attributes, e.g., *"three small red birds"*. Multiple attributes are concatenated with commas and conjunctions.
- **Action description**, which converts action-related attributes into short clauses, e.g., *"flying upward at accelerating speed"*, supporting both active and passive forms.
- **Background description**, which expresses environmental attributes (e.g., weather, lighting, time) as full sentences, e.g., *"The background is cloudy. The time is night."*

These components are concatenated to form the full prompt, for example: *"An image of three small red birds flying upward at accelerating speed. The background is cloudy. The time is night."*

An LLM (Qwen2.5-14B-Instruct (Yang et al., 2024)) is used only for minor grammatical corrections, such as:

- Pluralization (*"2 cat"* → *"two cats"*)
- Article insertion (*"red apple"* → *"a red apple"*)
- Verb tense consistency (*"is fly"* → *"is flying"*)
- Smoother conjunctions for multiple attributes

This two-step design ensures consistent and fluent prompts while avoiding uncontrolled expansion.

### A.4.2 MULTI-CHOICE QUESTIONS FOR AUTOMATIC EVALUATION

To evaluate generated images, we design a multi-choice question framework where both the entity and its attributes are verified by a multimodal large language model (MLLM). For each attribute or entity, up to five visually similar alternatives are sampled from the same subcategory, and two universal options ("Others" and "Can not answer") are appended. The first question always concerns the entity, and subsequent questions target attributes such as quantity, color, or background, each presented in a structured multi-choice format.

The MLLM is instructed to select the closest answers based on the image, while following strict rules: choosing "Can not answer" if the object is distorted or ambiguous, and choosing "Others" if none of the listed options match the visual evidence. In addition to entity recognition, this framework also enforces attribute-level consistency, with outputs represented as both a short natural language description and explicit multi-choice decisions. The generation success rate of a node is then defined as the proportion of answers that align with the intended entity and attribute labels. The evaluation of this method is presented in A.5.3.

### A.4.3 MODEL FOR SEARCH PRIORITIZATION

The accuracy predictor is designed as a lightweight transformer decoder that refines entity embeddings with attribute information. Each layer consists of self-attention on the entity embedding, cross-attention with attribute embeddings, and a feed-forward network, all connected through residual links and pre-layer normalization. Entities and attributes are first encoded with a pretrained text encoder, and missing attributes at shallower depths are padded with zeros to ensure consistent input structure. The final entity representation is passed through a two-layer feed-forward head with a sigmoid activation, producing a scalar in [0,1] that estimates the node's success rate. The model is trained with an L1 loss with the observed outcomes. During error slice discovery, the predictor is updated online with evaluation results and used to prioritize nodes predicted to have high failure likelihood, enabling efficient exploration under limited budgets.

## A.5 EXPERIMENTS DETAILS

### A.5.1 IMPLEMENTATION DETAILS

We apply FAILUREATLAS to a range of T2I models, including SD1.5 (Rombach et al., 2022), SDXL Turbo (Podell et al., 2023), SD3.5 Large Turbo (Stability AI, 2024), and Flux.1-dex (Labs, 2025). For all models, we adopt the hyper-parameters used in their official implementations (e.g., inference steps, guidance scale, and precision settings), shown in Table 3.

Table 3: Hyper-parameters used for different generative models. We adopt the default setting from official implementation.

| Model | Inference Steps | Guidance Scale | Precision |
|---|---|---|---|
| SD1.5 (Rombach et al., 2022) | 50 | 7.5 | FP16 |
| SDXL Turbo (Podell et al., 2023) | 1 | 0.0 | FP16 |
| SD3.5 Large Turbo (Stability AI, 2024) | 4 | 0.0 | BF16 |
| Flux.1-dex (Labs, 2025) | 50 | 3.5 | BF16 |

### A.5.2 VISUALIZATION AND CASE STUDY

We present six additional error slices for each model. For SD1.5, we observe that the model struggles with producing certain rare but plausible combinations, such as dusty chopstick or sitting zebra. It also often fails to render the racket or shuttlecock in badminton, frequently producing distorted images instead.

For SDXL Turbo, we find that it often ignores or simplifies control signals in favor of more common scenes. For instance, when prompted to generate a paper helicopter, it instead produces a paper airplane. Similarly, when asked for an oval-shaped cake, it outputs a circular cake.

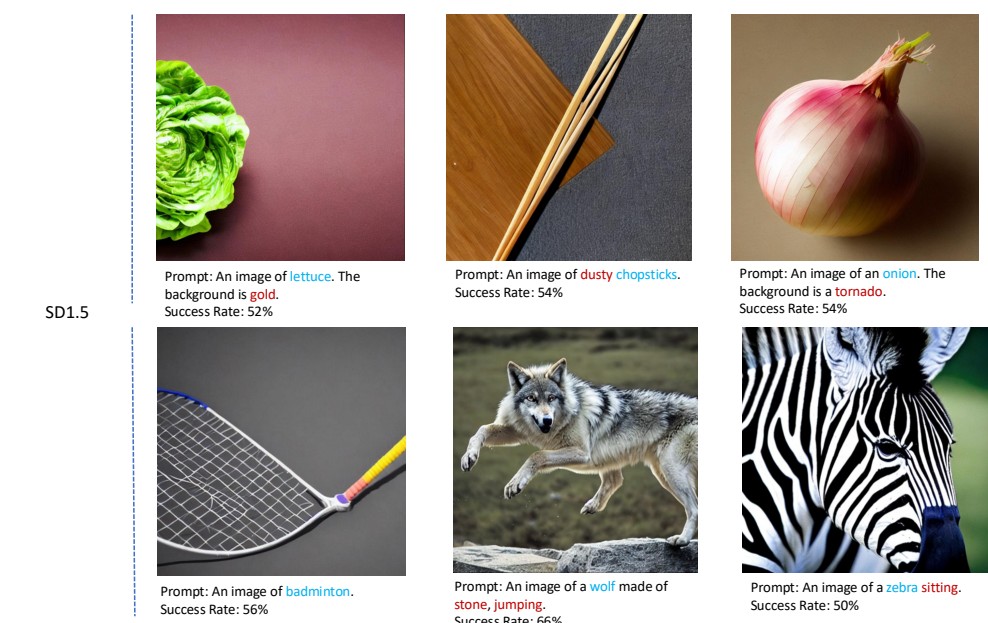

Figure 8: Example error slices of SD1.5. The prompt specifies entities in blue and attributes in red.

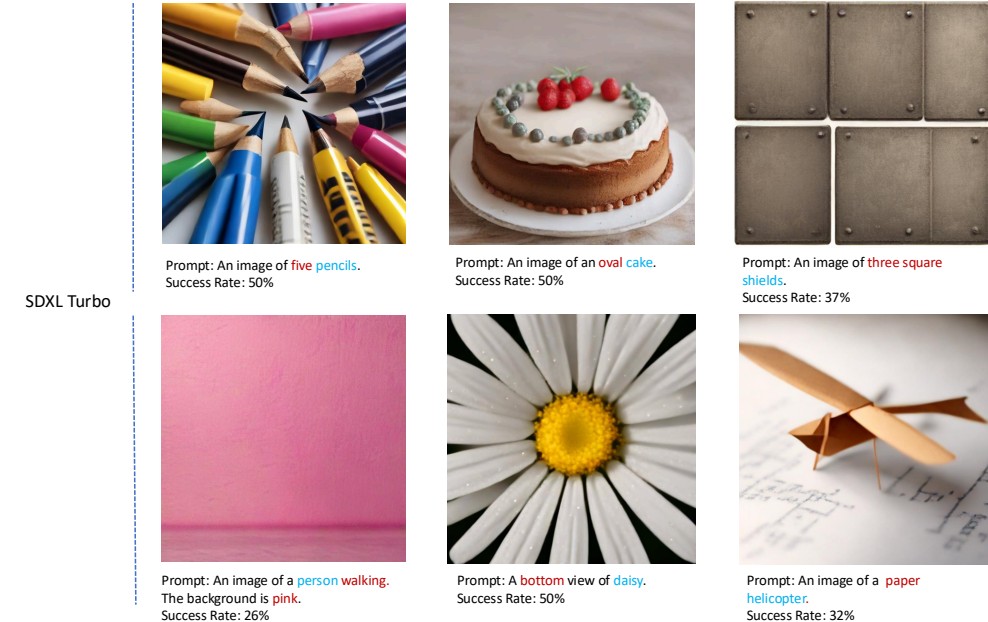

Figure 9: Example error slices of SDXL Turbo. The prompt specifies entities in blue and attributes in red.

For SD3.5 Large Turbo, we continue to observe failures in rendering certain entities, such as rubber bands. It also exhibits error patterns similar to earlier models. For example, prompting for a pink tomato often yields a red tomato, although other unusual colors such as purple can succeed. The model sometimes struggles with complex attribute controls, including lighting and artistic style, and frequently fails when asked to render a specific number of tools, such as scissors.

For Flux.1-Dev, we find failures in challenging scenarios such as bent keys or keys scattered horizontally. The model also struggles with rarer attribute specifications—for example, requesting a square camera rarely induces a shape change. Likewise, assigning the color gray to a camera often fails, even though other colors are consistently successful.

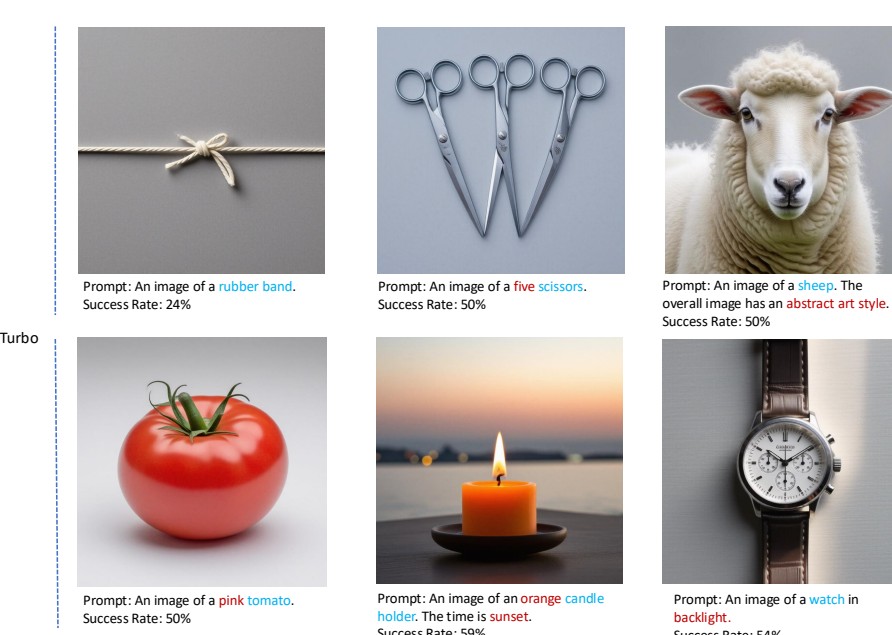

SD3.5
Large Turbo

Prompt: An image of a rubber band.
Success Rate: 24%

Prompt: An image of a five scissors.
Success Rate: 50%

Prompt: An image of a sheep. The overall image has an abstract art style.
Success Rate: 50%

Prompt: An image of a pink tomato.
Success Rate: 50%

Prompt: An image of an orange candle holder. The time is sunset.
Success Rate: 59%

Prompt: An image of a watch in backlight.
Success Rate: 54%

Figure 10: Example error slices of SD3.5 Large Turbo. The prompt specifies entities in blue and attributes in red.

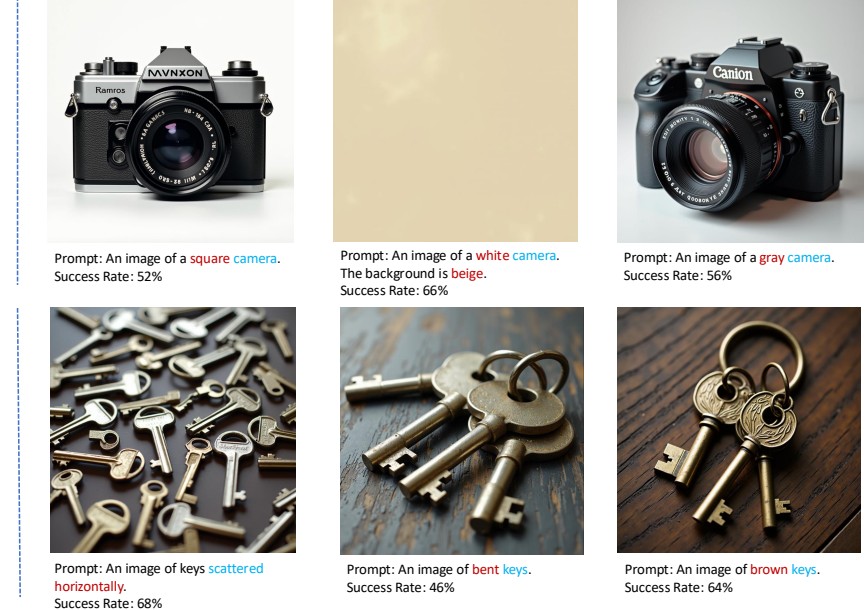

Flux.1 dev

Prompt: An image of a square camera.
Success Rate: 52%

Prompt: An image of a white camera. The background is beige.
Success Rate: 66%

Prompt: An image of a gray camera.
Success Rate: 56%

Prompt: An image of keys scattered horizontally.
Success Rate: 68%

Prompt: An image of bent keys.
Success Rate: 46%

Prompt: An image of brown keys.
Success Rate: 64%

Figure 11: Example error slices of Flux.1-dev. The prompt specifies entities in blue and attributes in red.

### A.5.3 BIAS ANALYSIS

To analyze the robustness of our evaluation, we examine two potential sources of bias. The first concerns the effect of repeated generations on the stability of node-level success rates. We sample 100K nodes for both SD1.5 and SDXL Turbo, and record the average change in success rate as the number of generations increases. As shown in Figure 12, the curve flattens after around 15 generations, and further sampling beyond 25 generations leads to an average change of less than 0.58% for SD1.5 and 0.46% for SDXL Turbo. This indicates that our choice of generating 25 images per node provides a sufficiently stable estimate of performance.

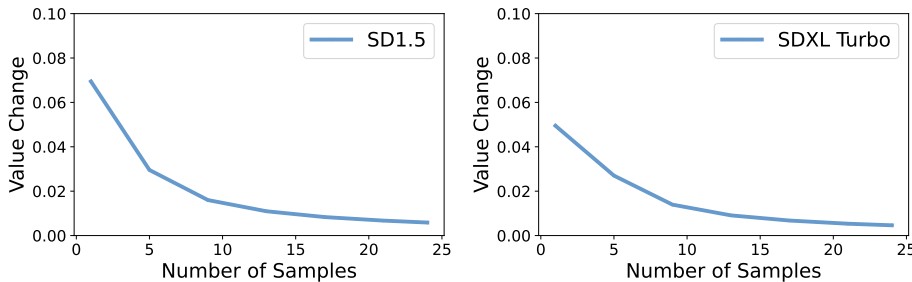

Figure 12: We analyze how the average success rate changes with the number of repeated generations. The success rate stabilizes with 25 generations, with additional samples leading to an average change of less than 0.0058 for SD1.5 and 0.0046 for SDXL Turbo.

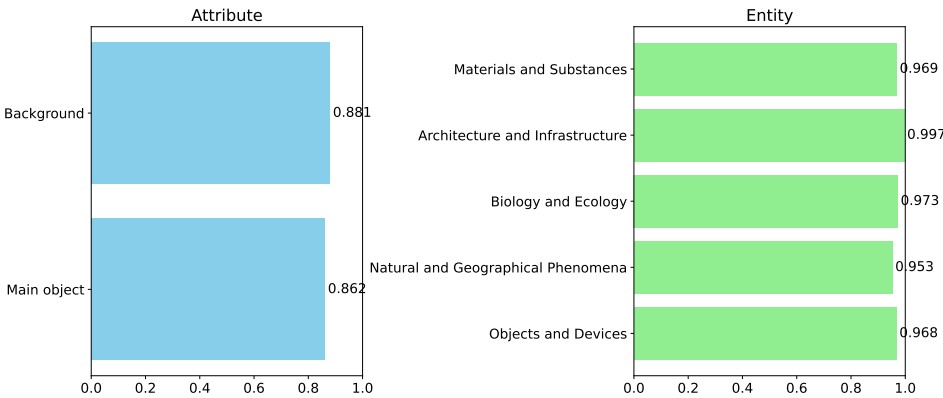

Figure 13: Human alignment test for the automatic evaluation method. Left: Results on attributes, grouped by categories. Right: Results on entities, grouped by categories.

The second potential bias arises from the evaluation procedure itself. To quantify this, we conduct a human alignment study: for each of the 758 entities and 437 attributes, we generate 5 images and compare model predictions with human judgments. Results are summarized in Figure 13, grouped at the category level. We observe high consistency, with an average alignment rate of 96% for entities and 86% for attributes. These results confirm that both repeated sampling and evaluation are stable, lending confidence to the reliability of our reported findings.

### A.5.4 TIME COST

As discussed in Section 5, the dominant computational cost arises from image generation and evaluation. Within the same search layer, both processes can be executed in parallel, and the overall runtime is therefore bounded by the slower of the two. For SD1.5, our framework explores 439,467 nodes, each with 25 image generations, resulting in approximately 11M generated images. The number of MLLM queries equals the number of nodes, adding further overhead. We acknowledge that this cost is substantial, but emphasize that it is mainly incurred for the purpose of exhaustively demonstrating the effectiveness of our method.

In practice, one often does not need to enumerate all possible error slices, but rather to identify a target number of them. In such cases, the prioritization strategy described in the main text enables significantly more efficient search. For example, in our experiments of SD1.5, at the third layer, discovering 10K error slices requires evaluating only about 12.5K nodes, while discovering 50K error slices requires about 65K nodes.

