# OpenReview forum: "FailureAtlas: Mapping the Failure Landscape of T2I Models via Active Exploration"
_ICLR.cc/2026/Conference — ICLR 2026 Conference Withdrawn Submission_

### Official Review · Reviewer_DNZB · 2025-10-28

**Soundness:** 2
**Presentation:** 2
**Contribution:** 2
**Rating:** 4
**Confidence:** 4

**Summary:**

This paper introduces FailureAtlas, a framework designed to systematically explore and map failure modes in text to image (T2I) generative models. The key idea is to move from passive benchmark evaluation to active exploration automatically probing a model’s capabilities through a structured entity attribute search.

FailureAtlas builds a large entity-attribute corpus and generates prompts through a tree search, and uses a LLM to automatically judge success or failure. To make the combinatorial search tractable two acceleration techniques were used a rule-based pruning and a learned prioritization predictor.

Experiments on several T2I models uncover hundreds of thousands of failure slices, many correlated with data scarcity in LAION-2B.

**Strengths:**

1. The idea of linking observed failures to data scarcity in the training corpus provides actionable insight for dataset design and model retraining.
2. The pruning and prediction based prioritization make the otherwise intractable search feasible, showing good design intuition and empirical justification.

**Weaknesses:**

1. Lack of comparative baselines: The paper doesn’t include any direct comparison with prior evaluation or diagnostic methods such as TIFA, GenEval. The only comparison shown is a vocabulary coverage table (Table 1), which says nothing about whether FailureAtlas actually finds more or better failures. There’s no evidence that the proposed framework improves diagnostic accuracy, interpretability, or coverage over existing tools. Without baselines, it’s impossible to know if this approach is genuinely better or just different.

2. Exploration scope: The search space focuses only on single-entity, attribute-level prompts. Real world T2I failures often involve more complex scenarios multiple entities, spatial reasoning, or contextual interactions which are out of scope here. As a result, the discovered failure slices mostly capture simple visual mismatches (like color or material errors) rather than compositional issues.

3. Correlation, not causation: The link between discovered failures and data scarcity in LAION-2B is only correlational. The paper doesn’t perform any controlled experiment (e.g., retraining with more examples) to show that scarcity causes the observed failures. This limits how strongly we can interpret the "data-driven insights."

4. The authors repeatedly claim that FailureAtlas is "the first active exploration framework for T2I models", but this is not accurate. The authors may not be aware but the broader concept of active failure exploration has been introduced before in works like HiBUG/HiBUG2 (Chen et al., 2024–25), FACTS (Yenamandra et al., ICCV 2023) and Failures Are Fated, But Can Be Faded (Sagar et al., ICML 2024). Those papers already framed model auditing as an active, structured search for failure cases. FailureAtlas mainly extends this paradigm to the T2I domain and scales it up, but it does not introduce a fundamentally new idea.

5. Ideally, the authors should compare FailureAtlas with other active or structured error-discovery. The current quantitative results are entirely self-referential and do not reveal how well this framework performs compared to prior work, or whether it adds practical diagnostic value.

**Questions:**

1. How does FailureAtlas compare to existing T2I evaluation frameworks (e.g., TIFA, GenEval which are mentioned in the paper) in terms of diagnostic coverage or failure-discovery rate? Do you have any quantitative or qualitative evidence showing that the discovered failures provide insights not captured by these benchmarks?

2. The evaluation depends entirely on Qwen2-VL-72B’s predictions.Have you tested the robustness of this automatic judge, for example, by cross-checking with other multimodal LLMs or a small human-annotated subset?

3. The pruning strategy assumes that generation difficulty monotonically increases as attributes are added. How robust is this assumption in practice? Are there examples where a more specific prompt (e.g., adding attributes) improves generation quality rather than causing failure?

4. The method defines failure when the generation success rate drops below 0.8. How was this threshold chosen?

---

### Official Review · Reviewer_D7ZP · 2025-11-01

**Soundness:** 2
**Presentation:** 2
**Contribution:** 3
**Rating:** 4
**Confidence:** 3

**Summary:**

The paper presents FailureAtlas, a framework that identifies failure cases in text-to-image (T2I) models by framing the problem as a structured graph search. The method systematically explores combinations of entities and attributes, locating parent nodes whose associated child nodes also tend to fail, thereby revealing broader, underlying weaknesses in the model. Through this process, the authors uncover a large number of failure patterns and provide insights into the model’s systematic shortcomings.

Overall, this is an interesting and valuable contribution that tackles an important problem in model evaluation and interpretability. However, in its current form, the paper suffers from several issues in clarity, organization, experimentation, and presentation that prevent me from fully endorsing it at this stage.

**Strengths:**

Framing the failure discovery process as a graph-search problem is both novel and interesting direction. This formulation enables systematic exploration of model weaknesses and facilitates structured reasoning about the underlying causes of failure. By analyzing parent–child node relationships, the approach helps reveal whether failures stem from broad conceptual gaps (e.g., difficulty with numerical reasoning in general) or from more specific cases tied to data scarcity or bias in the training distribution. Overall, the proposed framework offers a promising direction for diagnosing and understanding the limitations of text-to-image models.

**Weaknesses:**

**Major:**

1. Limited evaluation. The experimental analysis is restricted to SD 1.5 and SDXL, which are now relatively dated models. Evaluating on more recent and robust T2I systems would provide stronger evidence of the framework’s generality. In particular, studying how failure patterns evolve from older to newer architectures would offer valuable insights into model progress and persistence of error types.
2. Limited scope. Although the paper focuses on T2I models, the proposed graph-based exploration framework appears more broadly applicable. Similar hierarchical search structures could, in principle, be adapted for large language models or other generative systems. Demonstrating or even briefly discussing such extensions would considerably strengthen the paper.
3. Usefulness of identified failures. The paper stops short of exploring how the discovered failures could be leveraged for model improvement. A compelling ablation would be to test the transferability of failure slices. For instance, whether error cases identified in SD 1.5 persist in SDXL or newer models, thereby revealing which weaknesses are model-specific versus systematic.

**Minor:**

1. Presentation and space utilization. The manuscript still feels somewhat draft-like in presentation. For example, Figure 2 mainly shows data distribution statistics, which could be moved to the appendix since it contributes little to the main narrative. Figures 6 and 7 similarly occupy substantial space without offering enough analytical depth. Condensing or relocating such figures could improve readability.
2. Terminology ambiguity. The paper frequently uses the term “layer” to describe the levels of the search tree. This may be easily confused with the layers of the T2I model itself. Using an alternative term would help prevent misunderstanding.

**Questions:**

Asked in the form of weakness.

---

### Official Review · Reviewer_wiLp · 2025-11-01

**Soundness:** 2
**Presentation:** 3
**Contribution:** 3
**Rating:** 4
**Confidence:** 4

**Summary:**

This paper introduces FAILUREATLAS, a framework for actively exploring and mapping failure modes in text-to-image models at scale. The approach frames error discovery as a structured search for minimal failure-inducing concepts, and proposing two simplifying techniques to make this computationally challenging problem relatively more efficient. Applied to Stable Diffusion models, FAILUREATLAS uncovers a large set of error cases in SD1.5 and other T2I models, and establishes empirical connections between these failures and training data scarcity.

**Strengths:**

- The paper proposes a novel tool for active error discovery in T2I models that systematically reveals specific failure modes and limitations.
- The writing is clear and accessible, with main findings effectively communicated throughout the paper.

**Weaknesses:**

- The method is computationally intensive in its current form, requiring enumeration over large combinations of structured attributes and entities, albeit with approximations to reduce complexity. While the exploration is active, it operates within a static, pre-defined structure of attributes and entities—essentially a fixed search space.
- Although the paper frames FAILUREATLAS as a tool for active error discovery, the practical advantages over carefully designed evaluation benchmarks remain unclear. Since the method still involves enumerating possible entity-attribute combinations, it would be valuable to understand: (1) whether it discovers qualitatively novel error patterns not captured by systematic enumeration, and (2) whether it achieves more efficient discovery of unique errors under equivalent compute budgets compared to random or exhaustive sampling of entity-attribute combinations.

**Questions:**

- Is the proposed method sensitive enough to detect model-specific or training-specific differences, and can it reliably reflect these variations in the discovered failure patterns, e.g., what is the error overlap between different models, or between the same model trained on different datasets?
- How was the lightweight predictor trained? Specifically: (a) Does the method involve offline training, and if so, what training data was used? (b) If training occurs online during error discovery, is there sufficient data to train a reliable predictor, particularly at the very start of exploration/searching, will this make the predictor very unreliable, e.g., high l0 loss at small no. of explored nodes in Figure 5 middle,
- Does the tree structure, e.g., order of attribute layers (e.g., size → color vs color → size), affect the error discovered and findings? Additionally, how does the method handle cases where adding a deeper layer causes failure in an earlier attribute? For example, if "strawberry → large" generates correctly but "strawberry → large → red" produces a small red strawberry, the failure is attributable to the size attribute from the earlier layer, not the color attribute. How is error attribution handled in such cases?
- I am unclear about how Figure 5 right demonstrates that "the predictor achieves roughly a 2× speed-up in error discovery, enabling the identification of a large number of failures within limited search budgets." Could you explain what is being compared and how the 2× speed-up is quantified in this visualization?

---

### Official Review · Reviewer_Wq6d · 2025-11-03

**Soundness:** 2
**Presentation:** 2
**Contribution:** 1
**Rating:** 2
**Confidence:** 4

**Summary:**

The paper introduces an active exploration framework that composes prompts hierarchically from objects and attributes, then conducts a tree search to expose minimal failure‑inducing concepts in diffusion-based text-to-image generative models. To make tractable combinatorial search, it combines rule‑based pruning. This paper empirically applied to Stable Diffusion 1.5 and SDXL Turbo, and uncovers up to 440k error slices. By linking discovered slices to LAION‑2B, it shows many failures align with data scarcity, while others likely reflect data quality or inherent difficulty.

**Strengths:**

- **Interpretable diagnostics:** The hierarchical objects and attributes composition isolates minimal causes of failure rather than conflating factor.

- **Scalable exploration:** Pruning and prioritizer accurately reduce evaluations and speed discovery. For instance, retained layer‑3 nodes drop to 4.2% for SD1.5 and enables failure discovery faster.

- **Actionable link to data and reliable evaluation:** Data‑attribution connects many failures to low training frequency and the automated evaluator aligns well with humans.

**Weaknesses:**

- This paper doesn’t address timely topics that recent state-of-the-art diffusion models advanced at capturing user text prompts, particularly for objects and their properties. It not only adheres to models at the forefront, such as SD-v3.5 and flux, but its analysis relies on LAION-2B, released around three years ago. Considering the paper’s contribution, it doesn’t technically contribute to the domain of generative models.

- While this paper could be considered a benchmark, its focus on data point generation highlights how to make them. As a benchmark, it should justify the angles generative models are assessed and why they are meaningful. However, most of the paper’s content focuses on data point generation and efficiency of the procedure. I believe this approach would be beneficial if it could generalize across video generation or other types of text-based generative models. However, in its current form, it only considers text-to-image cases and doesn’t appear generalized to other types of models.

**Questions:**

As mentioned earlier, I was wondering if this approach could be applied to video generation tasks. I believe that the current video generation tasks still require relevant benchmarks to evaluate models based on their quality and ability to reflect user text prompts. I would greatly appreciate it if this approach could be extended to video generation.

---

### Note · Authors · 2025-11-14

I have read and agree with the venue's withdrawal policy on behalf of myself and my co-authors.